# Impact of Electrolyte Incorporation in Anodized Niobium on Its Resistive Switching

**DOI:** 10.3390/nano12050813

**Published:** 2022-02-28

**Authors:** Ivana Zrinski, Marvin Löfler, Janez Zavašnik, Claudia Cancellieri, Lars P. H. Jeurgens, Achim Walter Hassel, Andrei Ionut Mardare

**Affiliations:** 1Institute of Chemical Technology of Inorganic Materials, Johannes Kepler University Linz, Altenberger Str. 69, 4040 Linz, Austria; ivana.zrinski@jku.at (I.Z.); marvin.loefler@gmail.com (M.L.); achimwalter.hassel@jku.at (A.W.H.); 2Jožef Stefan Institute, Jamova c. 39, 1000 Ljubljana, Slovenia; janez.zavasnik@ijs.si; 3EMPA, Laboratory for Joining Technologies & Corrosion, Swiss Federal Laboratories for Materials Science and Technology, Überlandstrasse 129, 8600 Dübendorf, Switzerland; claudia.cancellieri@empa.ch (C.C.); lars.jeurgens@empa.ch (L.P.H.J.); 4Physics and Chemistry of Materials, Danube Private University, Steiner Landstrasse 124, 3500 Krems-Stein, Austria

**Keywords:** memristor, anodic oxide, niobium oxide, valve metals

## Abstract

The aim of this study was to develop memristors based on Nb_2_O_5_ grown by a simple and inexpensive electrochemical anodization process. It was confirmed that the electrolyte selection plays a crucial role in resistive switching due to electrolyte species incorporation in oxide, thus influencing the formation of conductive filaments. Anodic memristors grown in phosphate buffer showed improved electrical characteristics, while those formed in citrated buffer exhibited excellent memory capabilities. The chemical composition of oxides was successfully determined using HAXPES, while their phase composition and crystal structure with conductive filaments was assessed by TEM at the nanoscale. Overall, understanding the switching mechanism leads towards a wide range of possible applications for Nb memristors either as selector devices or nonvolatile memories.

## 1. Introduction

Volatility, scaling, speed and energy issues are the main obstacles in the field of complementary metal-oxide-semiconductor (CMOS) technology [1]. With advantages that include compatibility with CMOS and resistive instead of capacitive reading, memristors are frontrunners in the new generation of resistive random-access memory (RRAM) technology. High memory density, endurance, retention and low power consumption are some of the characteristics which can justify the possibility of memristors being used as an ultimate element for neuromorphic computing [2], in logic circuit applications [3], Recently, valve metals in their oxidized forms [4,5] have drawn scientific attention as memristive materials. The switching mechanism of memristive devices depends on the formation of conductive filaments (CFs) inside the insulating layer, which is the oxide sandwiched between two metallic electrodes [6]. The formation or rupture of CFs [7,8,9] is assisted by Joule heating and/or redox reactions leading towards reversible movements from cathode or anode interface of O ions, O vacancies, cations or electrolyte species [10]. These mechanisms are typically assigned to memristors showing unipolar or bipolar behavior [6,10]. The switching from high resistance state (HRS) to low resistance state (LRS) or vice versa from LRS to HRS (defined as SET or RESET processes, respectively) relies on voltage polarity for a bipolar memristive behavior [11]. In contrast, SET and RESET processes are independent of the voltage polarity for unipolar memristive switching [6,10]. Even though the coexistence of bipolar and unipolar switching was reported for several materials [12,13,14], a controllable switching [15] of anodic memristors on Nb in both regimes is demonstrated for the first time in this work. Besides, only several electrochemically grown Nb memristors have been studied recently, while their relevance for industrial implementation is emerging due to their simple, inexpensive and fast fabrication possibilities [16,17,18,19]. Hence, an influence of anodization of Nb thin films, used as bottom electrodes, in phosphate buffer (PB) or citrate buffer (CB) electrolytes was examined. The fabricated memristors are expected to have different behaviors given their active layers based on Nb_2_O_5_ [17,19], likely enriched with electrolyte species, as analyzed by both HAXPES and XPS. The study was motivated by the previous reports investigating the effect of tuning electrochemical parameters on anodic oxide growth [20,21,22,23] or memristive behavior, confirming the influence of electrolyte selection as the most relevant [24,25]. Since unipolar devices can be used for dense integration on diode selectors and bipolar ones show higher device lifetime [14], the controllable switching of the Nb anodic oxide memristors may be beneficial regarding the range of their possible applications [1,26,27,28,29]. Finally, the switching mechanism based on CF formation, influenced by their size and positioning, is suggested for devices showing reversible switching in unipolar and bipolar mode based on TEM investigation. This may lead towards the development of defect-engineered memristors applicable in memristive crossbars, for which modeling and theoretical support already exist. These models were recently developed in a PSpice Environment and LTSPICE Memristors Model Library for analysis of memory matrices with memristors based on transitional metals [30,31].

In the current work, the fabrication and analysis of Nb anodic memristors are described in Section 2, the structure of the memory layer is discussed in Section 3.1, and the compositional analysis of the memory layer is presented in Section 3.2. The performance of anodic memristors on Nb is given in Section 3.3, while their filamentary switching mechanism is analyzed in Section 3.4. The conclusions of the current work are presented in Section 4.

## 2. Materials and Methods

### 2.1. Fabrication of Anodic Niobium Oxide Memristors

The bottom electrode of the memristor was sputtered from a Nb target (99.95% Demaco Holland BV, Noord-Scharwoude, The Netherlands) in an ultra high vacuum system (Mantis Deposition, United Kingdom) on thermally oxidized Si wafers. The system base pressure was in the range of 10^−6^ Pa, while the sputtering was performed in low-pressure Ar plasma at 5 × 10^−1^ Pa. The intended thickness of the Nb thin film was 250 nm at the center of the wafer.

An oxide film was produced via electrochemical anodization in an electrochemical setup with Hg/Hg_2_SO_4_/sat. K_2_SO_4_ electrode (0 V vs. Hg/Hg_2_SO_4_ = 0.640 V vs. SHE) as reference electrode, a graphite foil matching the size of the Si wafer (99.8% Thermo Fisher Scientific GmbH, Dreieich, Germany) as counter electrode and the Si wafer with deposited thin film as the working electrode. In this way, formation kinetics and electric properties were controlled by a constant electric field, the anodization progressing in the high field regime [22]. The Nb thin films were potentiodynamically anodized at 8 V using a CompactStat potentiostat (Ivium Technologies BV, Eindhoven, The Netherlands) in order to grow the oxide up to a thickness of ≈20 nm. The cyclic voltammetry was performed by sweeping the potential from 0 to 8 V and backward from 8 to 0 V. The rate of potential increase was 100 mVs^−1^ for both 1 M PB and 0.1 M CB electrolytes, with final pH values of 7.0 and 6.0, respectively. Electrolyte solutions were prepared following standard preparation protocols [32] by mixing high-purity reagents (Na_2_HPO_4_, NaH_2_PO_4_, C_6_H_9_Na_3_O_9_, C_6_H_8_O_7_; Merck KGaA, Darmstadt, Germany) with ultrapure water (Arium mini, Sartorius, Göttingen, Germany). Commonly applied potentiostatic steps, performed at final potential values used during potentiodynamic anodizations, were skipped in this study for increasing the probabilities for defect formation and electrolyte incorporation, consequently [24,25,33].

Approximately 300 clusters, each containing 5 × 5 memristors, were finalized on each wafer by circular top electrode patterning (200 µm in diameter) through a Ni mask (Mecachimique, Pierrelaye, France). A detailed experimental procedure for the valve metal deposition by sputtering and Pt electrode patterning can be found elsewhere [25].

Memristive properties were investigated along the Si wafer by performing *I-U* sweeps, endurance and retention measurements using a Keithley 2450 SourceMeter and a self-developed gantry robot controlled via LabView software [25]. During all experiments, the Nb bottom electrode was contacted with a W needle and biased, while the top electrode remained grounded. The voltage values applied for *I-U* reached 2 V with current compliances up to 50 mA. The resistance values obtained during retention and endurance testing were measured by reading the current values corresponding to an applied voltage of 0.01 V.

### 2.2. Characterization and Analysis Methods

Impedance spectroscopy (IS) measurements were conducted on memristors grown in CB or PB using a potentiostat (CompactStat, Ivium Technologies BV, Eindhoven, The Netherlands). The measurements were performed in 2-electrode mode at 0 V bias applying an AC excitation signal with an amplitude of 100 mV. The frequency was swept between 100 kHz and 1 Hz.

The crystal structure and phase composition of the memristor samples were assessed by transmission electron microscope (TEM, JEM-2100, JEOL Inc., Tokyo, Japan) operating at 200 kV. A double-tilt analytical Be holder was used for micromanipulation of the thin sample. Micrographs were recorded by a slow-scan CCD camera (Orius SC1000, Gatan Inc., Pleasanton, USA). Scanning transmission electron microscopy (STEM) experiments were conducted by field-emission TEM (S/TEM, JEM-2010F, JEOL Inc., Tokyo, Japan), operating at 200 kV and additionally equipped with an energy-dispersive X-ray spectrometer (EDS, Link Pentafet Mod. 6498, Oxford Instruments Ltd., Abingdon, UK). The high-angle annular dark-field (HAADF-STEM) micrographs were recorded by an annular detector using a probe at a semiangle of 17 mrad and a 35~90 mrad detector.

Electron-transparent samples for TEM analyses were prepared by site-specific focused ion beam (FIB, Helios NanoLab 650i, FEI BV, Eindhoven, The Netherlands) with Ga ion source. Rough milling was conducted at 30 kV, 2.5 nA, while during step-by-step thinning of the lamella, the current was reduced from 2.5 nA to 0.79, 0.43 and 0.23 nA, while final polishing was done at a high tilt at 80 pA.

The chemical composition of anodic memristors grown in different electrolytes was evaluated by X-ray photoelectron spectroscopy (XPS) and hard X-ray photoelectron spectroscopy (HAXPES). Analyses using HAXPES and XPS were performed with a PHI Quantes spectrometer (ULVAC-PHI) equipped with a conventional low-energy Al-Kα X-ray source (1486.6 eV) and a high energy Cr-Kα (5414.7 eV) X-ray source. Both sources are high-flux focused monochromatic X-ray beams, which can be scanned across a selected area on the sample surface. The energy scale of the hemispherical analyzer was calibrated according to ISO 15472 by referencing the Au 4f7/2 and Cu 2p3/2 main peaks (as measured in situ for corresponding sputter-cleaned, high-purity metal references) to the recommended binding energy (BE) positions of 83.96 and 932.62 eV, respectively. Charge neutralization during each measurement cycle was accomplished by a dual-beam charge neutralization system, employing low-energy electron and Ar ion beams (1 V bias, 20 μA current). The core level and Auger transition measurements with both sources were performed in high-power mode, adopting an elliptical surface analysis area with a long axis of roughly 1400 μm. The step size and pass energy for core level measurements were 0.13 and 69 eV, respectively, for both sources (surveys were acquired at 280 eV pass energy and 0.5 eV step). The atomic concentrations were calculated from the peak areas after Shirley background subtraction using the predefined sensitivity factors in the MultiPak 9.9 software provided by ULVAC-PHI. The binding energy scale was corrected by aligning the lower edge of the upper valence band of the oxides to that of the Nb metal with a native oxide.

## 3. Results

### 3.1. Nanoscale Structure of Niobium Oxide Anodic Memristors

An overall view of the memristive structure is presented in Figure 1a. The Si substrate and the thermally grown amorphous SiO_2_ are observable, the latter with a measured thickness of ≈215 nm. Furthermore, Nb thin films were 220 nm thick and composed of columnar Nb crystals extending through the whole thickness of the film, characteristic of sputter deposition. The top layer of Nb thin film has a well-developed morphology with exposed {100} and {110} planes and is covered by an amorphous Nb oxide film, on average 20 nm thick. The memristive structure is topped by sputtered deposited ≈ 240 nm of Pt. During FIB sample preparation, the multilayer stack was additionally coated with an electron-deposited Pt layer (e-Pt), followed by an ion-deposited thicker Pt film. The analysis of selected area electron diffraction (SAED) patterns (Figure 1b,c) indicated that the Nb oxide film is amorphous, without any crystalline phases. Diffractograms recorded in the Nb bottom and Pt top electrodes can be easily compared to the diffractogram recorded in the center of the device, where the memristive anodic oxide resides. In each case, simulated and experimental curves match well, and no additional crystalline phases, i.e., belonging to Nb_2_O_5_, were observed.

The distribution of the principal chemical components, Nb, Pt and O, was first probed by TEM-based EDS, and the results are summarized in Figure 2. First, a suitable region of the sample was identified where interfaces between Nb bottom electrode, amorphous anodic Nb oxide and Pt top electrode are more or less parallel to the electron beam, then a series of consecutive point spectra were recorded (Figure 2a). The EDS acquisition region coincides with the X-axis of the graph, with the probe diameter of approximately 2 nm. Due to overlapping of the most intense Nb-L and Pt-M energy lines between 2 and 2.2 keV, the characteristic Pt-Lα (9.4 keV) and Nb-Kα (16.6 keV) energy lines were considered for calculating the relative concentrations of the Nb, O and Pt, according to the Cliff–Lorimer ratio method [34]. As it can be seen, the transition between Nb and Nb oxide is gradual, which can be attributed to the ingress of oxide phase into the grain boundaries between Nb columns and uneven development of the Nb oxide on the exposed Nb crystal surfaces. The interface between Nb oxide and Pt film is sharp and well defined, following the morphology of the underlying Nb. To further explore the possible incorporation of electrolyte species, a point EDS analysis at multiple positions inside the amorphous anodic Nb oxide was additionally performed (Figure 2b). The Nb–O ratio changes in depth and the presence of substoichiometric oxides were suggested in the proximity of the Nb interface, for both cases of anodization in different electrolytes. Besides Nb and O, no other elements were detected in the amorphous Nb oxide layer by using this method. The (Cu) peak is an artifact originating from the support grid.

### 3.2. Anodic Memristors XPS/HAXPES Analysis

The composition of anodic oxides grown in PB and CB electrolytes was analyzed by XPS. In Figure 3, the XPS surveys of two samples grown in different electrolytes (acquired with Al source) are shown. The incorporation of P in the PB-grown sample and of Na in the CB-grown sample is evidenced. The nominal chemical composition of the surface region of each oxide film, as determined by XPS, is also reported in Figure 3. As evidenced by the XPS sputter depth profiles shown in the Appendix A, the concentration of Na and P in the interior of the oxide films are well below the detection limit (i.e., lower than about 0.1–1 at.%). Hence, Na and P are present only at the surface, i.e., in the vicinity of the Pt top electrode.

The presence of different electrolyte species (P and Na) in the PB and CB oxide films may result in a different chemical environment of Nb in the oxide film and thereby may trigger a different memristive switching behavior. To investigate this in more detail, the Nb 3d core level region was measured at different probing depths using either the Al-Kα or Cr-Kα source, as shown in Figure 4. The probing depth of Nb 3d photoelectrons emitted using soft Al-Kα radiation is about 3 nm, while the corresponding probing depth with hard Cr-Kα radiation is around 20 nm. As reflected in Figure 4, the binding energy (BE) of the Nb 3d main peak is identical for both probing depths, whereas the PB oxide film experiences a BE shift of more than 0.5 eV when increasing the probing depth from 3 to 20 nm. Notably, dual-beam charge neutralization was applied during the measurements, and the BE of the C 1s main peak is unaffected when shifting between the Al and Cr source. The results, therefore, suggest that the oxide films grown in CB are more homogeneous in composition than those grown in PB. The BE shift of the Nd 3d with increasing probing depth for the PB oxide film may be caused by different oxide stoichiometries at the surface and in the bulk of the oxide. Namely, a Nb 3d BE position of about 207 eV is characteristic of Nb_2_O_5_, whereas a respective BE position of about 206 eV is indicative of NbO_2_ [35]. Hence, the CB oxide films seem to have a constant composition up to the surface (i.e., Nb_2_O_5_), whereas the composition of the PB oxide film deviates at the surface (i.e., Nb_2_O_5_ at the surface and NbO_2_ towards the bulk).

To investigate the origin of the depth-dependent Nb 3d BE position for the PB-grown oxide film, a local chemical state analysis of Nb and O in the grown oxides was performed by comparing the depth-dependence of the corresponding Auger parameter (AP) values. The AP value of a given atom in the solid, as originally proposed by Wagner, is defined as the sum of the BE of a strong core-level photoelectron line and the kinetic energy (KE) of a respective prominent and sharp core–core Auger transition [36]. Notably, the AP value is independent of static charging and shifts of the Fermi level due to, e.g., band bending effects in thin films. The AP value of a given atom in the solid can be directly related to the local electronic polarizability of the solid upon core-hole formation, which is very sensitive to structural changes in the nearest coordination sphere of the core-ionized atom [37]. A representation of the AP of a given atom is obtained by plotting the kinetic energy of the Auger transition on the ordinate and the binding energy of the photoelectron line on the abscissa (in the negative direction). Lines of constant AP values are represented by a diagonal grid of slope 1 [38]. Accordingly, the AP values of O (α_O_) in the oxides were determined by acquiring the O 1s photoelectron and O KLL Auger lines, as measured with the Al-Kα source. The corresponding Nb AP values (α_Nb_) were derived from the Nb 3p photoelectron line, as measured with the Al-Kα source, and the Nb LMM Auger line, as measured with the Cr-Kα source. The photon sources for measuring the combinations of photoelectron and Auger lines were chosen to obtain a similar probing depth of roughly ~5 nm. In this way, the chemical state analysis of the anions and cations can be carried out at a similar probing depth, thus avoiding artifacts related to in-depth inhomogeneities. The obtained Wagner plots for O and Nb in the grown oxide films are shown in Figure 5a,b.

As it is evident from Figure 5a, both oxide films have the same O Auger parameter (α = 1041.8 eV), indicating a similar local chemical environment of O anions. On the contrary, the local chemical state of Nb differs for the CB and PB oxide films (see Figure 5b). The PB oxide films have a larger AP than the CB oxide film (i.e., the AP shifts towards that of the pure Nb metal, as indicated with a star), which indicates that core-ionized Nb cations are more effectively screened by valence electrons in the PB oxide. The reduced screening efficiency of valence electrons in the CB oxide could originate from an increase in the oxide band gap, resulting in a lower electronic polarizability around the core-ionized atom (as compared to the PB oxide). Additionally, it can be inferred that oxides formed in CB are more uniform and compact than those formed in PB due to the fact that P requires more space, likely randomizing the position of neighboring atoms or molecules. A similar trend was observed in recently studied amorphous Ta oxide growing in PB and CB and having the same stoichiometry of the oxide as in the current work [24]. Therefore, the selection of the electrolyte may be relevant regarding the electrical characteristics of memristors due to the compositional homogeneity and compactness of the samples grown in different buffers.

### 3.3. Electrical Behavior of Niobium Oxide Anodic Memristors

The resistance values of Nb_2_O_5_ memristors in their initial state were in the range of 10^4^ Ω, indicating modest semiconducting properties, as already reported [39]. A reproducible switching between HRS (≈10^4^ Ω) and LRS (≈10^2^ Ω) was established via an electroforming process in a negative direction for devices anodized in PB (up to −3.5 V) and in a positive direction for devices formed in CB (up to 2.5 V). When devices were biased by higher voltage values, they were irreversibly switched to LRS. The switching behavior of both anodic memristors was extracted from *I-U* sweeps as indicated in Figure 6. Theoretical descriptions of memristive effects in similar oxides were recently modeled based on the Lehtonen–Laiho approach [30,31]. As it can be seen in Figure 6, memristors switched in lower current compliance regime (up to 1 mA) showed unipolar behavior (Figure 6a,b,d) as opposed to bipolar memristors recorded with current compliance in the range from 5 to 50 mA (Figure 6a,d,e). Unipolar and bipolar switching modes were always dependent on the current limitation values *I_cc_* for devices formed in both CB and PB (Figure 6a,d). It is relevant to note that unipolar memristors (Figure 6b) operated in a high current compliance regime (or without it) resulted in irreversible LRS. However, devices showing bipolar behavior (Figure 6e) can be safely switched at negative voltage values without current limitations (*I_cc_* was set but not reached) or with high current compliances at both polarities.

The switching voltages were dependent on the electrolyte used for anodization. For devices anodized in PB, the voltage values decreased with higher current compliances, whereas the voltage values increased with the current limitation for devices formed in CB, as expected. Since devices formed in PB reached maximum switching range up to ±1.5 V, it may be assumed that unipolar devices anodized in PB will consume more power in the low current compliance regime.

It should be also noted that devices anodized in PB and CB demonstrated both threshold and nonvolatile characteristics. Only after the electroforming process, devices were switched from initially HRS to LRS once the applied voltage reached a threshold value. Following that, the current increased to its maximum or compliance value, thus allowing devices to switch to LRS. In contrast, HRS was reached at reduced voltage values, hence hold voltage. This can be seen in Figure 6a. The switching mechanism of both unipolar and bipolar devices is discussed further in detail.

In order to study multi-level switching, LRS was extracted from *I-U* sweeps for memristors anodized in PB and CB (Figure 6c,f). The error was calculated based on the results for at least 25 different memristors recorded at each current compliance. Clearly, devices formed in PB showed up to four distinguishable switching levels, and memristors formed in CB showed even up to eight switching levels. In both cases, LRS values linearly increased with the decrease in the current compliance *I_cc_.* Hence, devices formed in CB will more likely store more than one bit per cell, which is an important memory characteristic for neuromorphic applications [4,28,40]. The reduced electron screening in oxides grown in CB, as discussed before in the HAXPSE analysis, could rationalize their improved memory characteristics. This is also in agreement with previous studies in which Hf-Ta-based oxide memristors with a lower electronic polarizability also exhibited lower cycle-to-cycle variabilities [41].

Furthermore, HRS/LRS ratio, as an important electrical characteristic, was examined using not only *I-U* sweeps but also endurance and retention measurements for 5–25 memristors recorded for both groups of samples anodized in PB and CB (see Figure 7). Generally, LRS values ranging from 100 Ω to 1 kΩ and HRS ranging from 10^4^ to 10^7^ Ω gave HRS/LRS higher than 100 Ω. Cycle-to-cycle variability for LRS and HRS during endurance or retention tests recorded for different devices was empirically determined and is shown within blue and red confidence bands, respectively. The range of resistance state ratio values is comparable with literature values obtained for anodic memristors based on different valve metals [18,25,42]. Memristive devices on Nb formed in PB and CB showed high HRS/LRS ratios and low leakage currents. The strongest memristive effect can be observed for devices with the strongest hysteresis loop obtained for *I-U* sweeps which were recorded with higher current compliances *Icc* (Figure 6a,d).

The exemplifying *I-U* curves measured with a limitation of 5 mA for a number of cycles during consecutive writing processes are shown in Figure 5b. Low resistance state values showed instabilities, whereas HRS became highly conductive, suddenly ending the lifetime of devices anodized in PB after 10^6^ cycles. In contrast, memristors formed in CB demonstrated stable LRS and HRS with equally long lifetime but higher device-to-device variability. A similar trend is observed for the reading procedures, but instabilities were recognized for HRS when both electrolytes were used for anodization. In the case when CB was used, cycle-to-cycle variability of HRS was higher for 25 different devices (Figure 7b,d). Hence, it can be expected that electrolyte selection may play a crucial role in memristive performance. Furthermore, endurance and retention of the devices switching either in bipolar or unipolar mode were also compared. The lifetime of unipolarly switched devices was shorter, reaching approximately 1000 cycles (data not shown here). This is in agreement with literature reports claiming excellent endurance for bipolar devices [14].

The behavior of resistive states was investigated by impedance spectroscopy (IS). The Bode representations of IS measurements for anodic Nb memristors (shown as Appendix A) show very similar behaviors regardless of the buffer used for anodic oxide formation. In all cases, in the LRS, the memristors behave as a pure resistor, indicated by a low resistance with a phase shift close to 0. In the HRS, a dominant capacitive behavior can always be concluded due to the phase shift approaching −90° at high frequencies. This may be expected since both LRS and HRS are easily distinguishable, which is clearly visible from the high opening of the hysteresis loop of *I-U* curves recorded for devices grown in PB and CB. Fitting the IS data with the measured thickness of oxide grown in CB and PB based on TEM allowed calculating the memristor capacitances and relative permittivities of oxides grown in PB and CB as 130 and 120, respectively, in good agreement with the literature [43].

Taking into account all observed memristive characteristics, the switching mechanism can be interpreted. It is already well known that after the formation process, LRS and HRS switching is accomplished by recombination of O vacancies with O migrating due to the electric field, thus forming CFs inside the oxide [10]. However, the recombination slightly differs for unipolar and bipolar modes. Generally, the switching to HRS is based on the voltage-controlled mobile O and O vacancy movement along defect-enriched regions (i.e., grain boundaries, dislocations) from cathode to anode interface, where electrochemical oxidation of CFs occurs [10,14]. It should be noted that O vacancy movement mainly refers to crystal lattices, as opposed to amorphous systems such as Nb_2_O_5_ which can be correlated with O migration [44,45,46,47]. Thus, once the voltage value of opposite polarity is applied, O or electrolyte species will drift back to the cathode interface where the reduction of CFs will take place, and thus switching to LRS will be set [14]. In this work, devices formed in PB were set to LRS by switching in a negative direction until the threshold voltage was reached. The current compliance was not necessary, suggesting that no thermal disturbance was present. By applying positive voltage values, the devices were switched to HRS at the hold voltage value. Memristors formed in CB followed the same mechanism, but opposite voltage values were applied (as compared to memristors formed in PB) in order to switch to LRS and HRS. In the case of unipolar devices, the switching to LRS was set by reaching the threshold voltage of positive polarity. The switching to HRS was obtained when the voltage was reduced up to the hold voltage of the same positive polarity. The *I-U* sweeps did not exhibit switching at both polarities, as opposed to bipolarly switched devices, in which the opening of the hysteresis loop is visible at both sides (Figure 6a,d).

As already mentioned, current limitation *Icc* is required in the case of unipolar operating mode. This can be explained by the switching mechanism in which defect-enriched regions are generated due to thermally assisted redox reactions. A local reduction/oxidation, driven by Joule heating and applied electric field, leads towards CF formation/rupture [10,14]. If the current is not limited, CFs regions can be thermally damaged, promoting the irreversible dielectric breakdown of Nb_2_O_5_.

Anodic memristors on Nb have shown the coexistence of unipolar and bipolar switching behavior in the same device. By understanding the described mechanisms, it may be assumed that CF formation is controlled by the Joule effect when switching in a low current compliance regime. When increasing the current limitation, a migration of mobile O and electrolyte species will take over the switching process. Despite the fact that memristors operating in bipolar mode do not always reach the current compliance, an upper limit is defined for 50 mA, the value at which the Joule effect synergizes with the uncontrolled drift of species inside the oxide. Depending on the application of memristors, an operating mode can be selected by adjusting the current compliance. It should be also emphasized that devices are switched reversibly from unipolar to bipolar mode. Additionally, devices exhibited threshold and nonvolatile characteristics which can broaden their application spectra as selector devices and memory elements in one structure, thus avoiding the incorporation of several devices in one crossbar array [27].

Finally, a mixed behavior will benefit endurance and retention of memristive devices. Until now, only several Nb anodic memristors reaching a maximum of 1000 switching cycles had been fabricated [17,18,19]. Limited endurance and retention performance occurs due to electrical and thermal disturbances through fragile CFs or the change in their shape and position after each cycle [14,48]. The anodic memristors developed in the present study can undergo more than 10^6^ switching cycles, which is attributed to the formation of stable CFs controlled by the Joule effect and bipolar switching dictated by the movement of O and/or electrolyte species (P or carboxyl groups) incorporated in Nb_2_O_5_. Additionally, the coexistence of unipolar and bipolar switching behavior makes memristive performance improved in a way that thermal damage of CFs typical for unipolar mode can be avoided by simply selecting the consecutive switching in bipolar mode when necessary. In addition, simultaneous movement of electrolyte species and mobile Nb cations through CFs defined by redox reactions makes bipolarly switched memristors less prone to unpredictable switching due to forming CFs at different positions after each cycle, which is a case for unipolar devices [14]. Similarly, the reading of the resistance state cannot be affected by nondistributed CFs, keeping the HRS/LRS ratio constant (Figure 7b,d). On the one hand, the irreversible change of resistance to HRS (clearly visible in Figure 7d for memristors grown in CB) may be correlated to a phase segregation process, in which the number of metallic Nb-rich regions decreases, being progressively isolated by the local increase in the O in the matrix. On the other hand, the end of device lifetime may be also justified by the formation of O-enriched gaps in the CFs at the interfaces of the electrodes, thus leading to permanent switching to HRS during consecutive pulsing [49]. However, the top electrode material is crucial for defining the maximum number of switching cycles in bipolar devices since O migrates from the interface to the bulk, thus forming O reservoirs near Pt interface accumulating after each set to HRS. This may lead towards the sudden O diffusion into CFs and eventually the end of the device lifetime [10,48]. Such an event is evident for memristors with oxide grown in PB (Figure 7a,c). Bipolar switching mode is highly favorable for anodic memristors on Nb and can be easily selected by choosing the current compliance range, thus influencing the overall electrical performance of devices.

### 3.4. TEM Analysis of Conductive Filaments

The anodic oxide grown in CB appeared more compact and uniform, thus increasing the probability of more concurrent CF formation during the electroforming process, responsible for the switching at more resistive levels. In a previous study, the CF pinning was dictated by the P incorporation forming a new substance in oxide [24]. Nevertheless, the size of P and its atomic interactions may prevent the formation of a high number of fully developed (i.e., connecting both electrodes) concurrent CFs. Instead, this will support only several CFs or even one CF positioned according to P, thus limiting the multilevel switching behavior. The phase-contrast TEM (HR-TEM) micrographs of the compact structure of amorphous oxide grown in CB as well as CFs imaged in the oxide grown in PB are shown in Figure 8. The selected images are representative of each sample. Each set of figures is composed of original HR-TEM micrographs with marked Pt and Nb crystal planes, as well as outlined CFs (orange lines) in the Nb oxide. On the right side, the same HR-TEM images are presented with contrast histogram variations (marked in inset) to enhance the features in the Nb oxide layer. The CFs were determined on these images. Blue squares mark the region used for fast Fourier transform (FFT; no periodic structures, i.e., crystallites, were detected) and average background subtraction filtered (ABS-f) insets.

Since the conductive filaments are formed by O vacancies, their TEM visualization is extremely difficult. The CFs are one-dimensional structures, embedded in ≈100 nm of amorphous, randomly distributed Nb oxide clusters. Due to the size of the CFs, the HR-TEM seems to be the method of choice for their visualization. In HR-TEM, the contrast in the micrographs is formed due to the differences in the electron wave phases. These are caused by specimen interaction, mainly due to the differences in the electron densities in the image plane. Scattering of the incident beam by the sample leads to a change in the amplitude and phase of the electron waves, which results in amplitude contrast and phase contrast, respectively. While phase contrast is mainly used for visualization of periodic structures (i.e., lattice imaging, where direct and diffracted beams undergo phase shifts while passing through the material and the image is formed by recombination of all the beams into a resulting interference pattern), the mass–thickness contrast is the primary contrast source in amorphous materials [34]. In the present case, Nb oxide is amorphous, and hence the diffraction contrast is absent, and additionally, the thickness of the sample is uniform due to FIB lamella preparation.

Based on these assumptions, it is safe to conclude that the CFs, formed as the redistribution of O vacancies in the strong electric field, bring only a small contribution to the contrast variation, which manifests as a deviation from the otherwise homogeneous appearance of the amorphous oxide matrix. Such an example can be seen in Figure 8, where on thresholded micrographs the CFs can be visualized as black or white linear structures, emerging from metal electrodes. As the human eye cannot see intensity differences smaller than 5–10%, a digitally enhanced contrast is performed by selecting only the corresponding portion of the histogram.

## 4. Conclusions

Anodic memristors grown in PB and CB showed both nonvolatile and threshold memristive switching behaviors. The selection of the electrolyte used for the anodization strongly influenced the electrical and memory characteristics of the memristive devices. Memristors formed in CB exhibited multilevel switching characteristics, while devices grown in PB showed lower cycle-to-cycle and device-to-device variabilities obtained during endurance measurements. Generally, devices grown either in PB or CB could be reversibly switched in bipolar and unipolar mode. The desired switching mode can be selected by adjusting the current compliance, which can broaden the application spectra of niobia devices.

## Figures and Tables

**Figure 1 nanomaterials-12-00813-f001:**
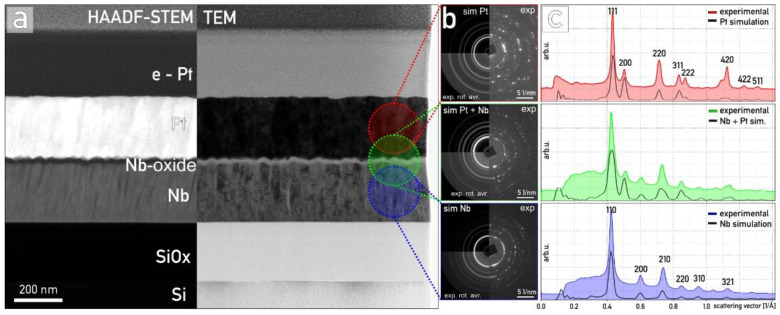
The general composition and layer stack of the samples (here, the sample anodized in CB): (**a**) HAADF-STEM and TEM overview composite micrograph shown with marked main features. (**b**) The representative SAED patterns recorded from marked regions are presented as raw data, rotationally averaged and simulated patterns for pure Pt, pure Nb and Pt and Nb mixture. (**c**) Intensity profiles for each SAED are compared to simulated patterns.

**Figure 2 nanomaterials-12-00813-f002:**
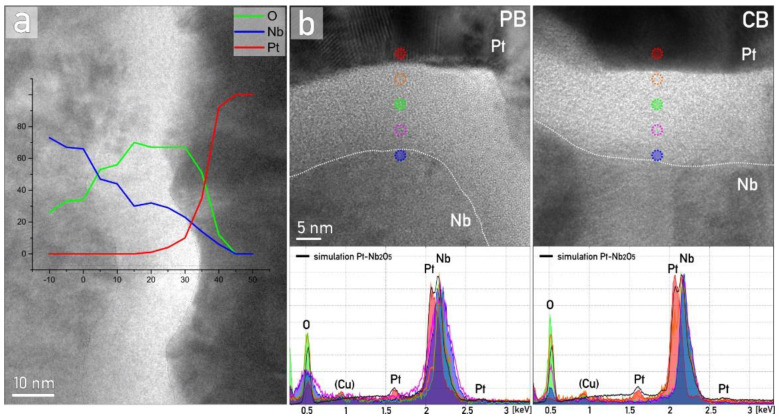
(**a**) TEM micrograph and EDS line profile through the interface between Nb, Nb oxide and Pt for CB sample. (**b**) HR-TEM and EDS spot analyses at various locations of Nb oxide of PB and CB samples, with the representative part of experimental spectra between 0.3 and 3.3 keV and simulated Pt-Nb_2_O_5_-Nb spectra for comparison (black line). The position of EDS probe analysis is marked and color-coded to correspond with EDS spectra.

**Figure 3 nanomaterials-12-00813-f003:**
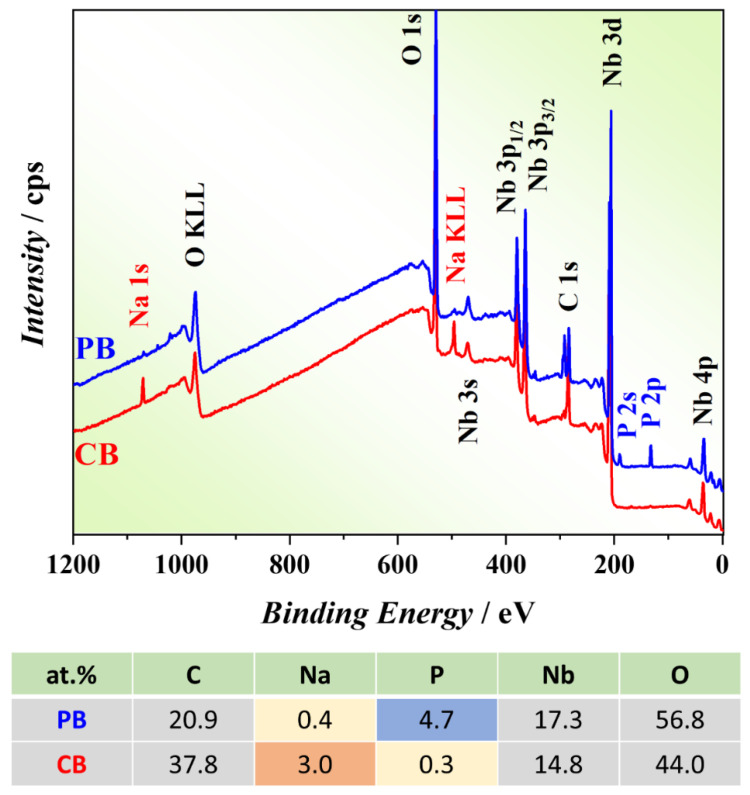
XPS survey spectra acquired for PB- and CB-grown samples and the derived nominal composition.

**Figure 4 nanomaterials-12-00813-f004:**
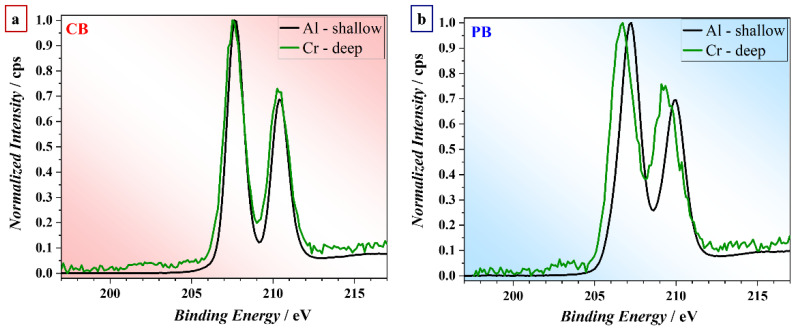
The Nb 3d_5/2_ region, as measured with the soft Al and hard Cr sources for the (**a**) CB and (**b**) PB samples.

**Figure 5 nanomaterials-12-00813-f005:**
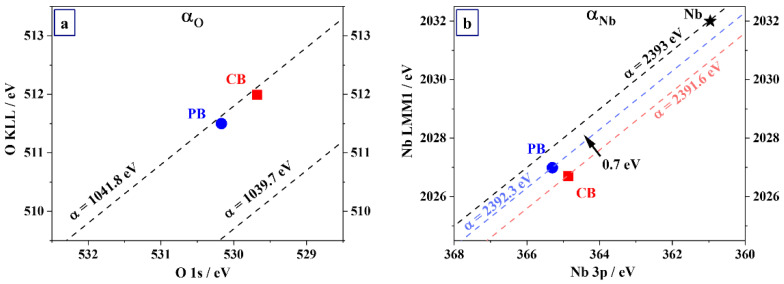
Wagner plots of (**a**) O and of (**b**) Nb in the grown oxide films. A difference in chemical state of Nb for the oxide films grown in PB and CB electrolytes is evidenced. The α for the Nb metallic, as indicated with a star, was acquired on a 300 nm PVD Nb layer (after removal of the native oxide by sputter cleaning).

**Figure 6 nanomaterials-12-00813-f006:**
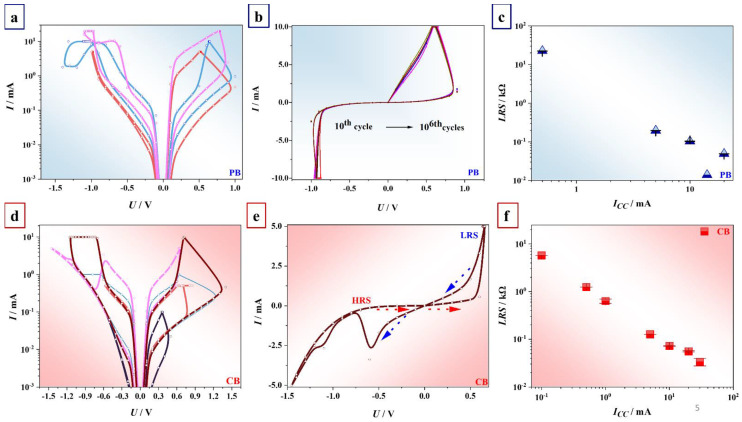
(**a**,**b**) *I-U* sweeps recorded for memristors grown in phosphate buffer solution (PB). (**c**) Low resistance state values (LRS) for memristors grown in phosphate buffer.(PB) (**d**,**e**) *I-U* sweeps recorded for memristors grown in citrate buffer solution (CB) with switching direction to high resistance state(HRS) and low resistance state (LRS). (**f**) Low resistance state values LRS for memristors grown in citrate buffer (CB).

**Figure 7 nanomaterials-12-00813-f007:**
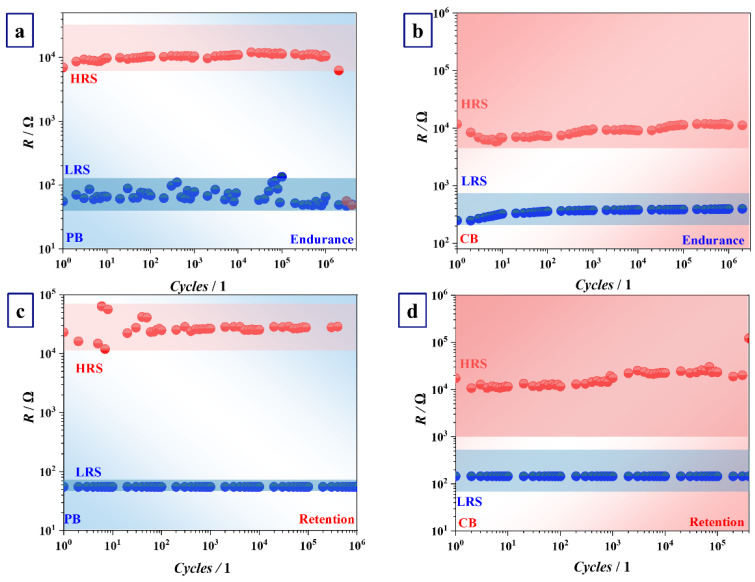
Endurance and retention of memristors grown in phosphate (**a**,**c**), and citrate buffers (**b**,**d**).

**Figure 8 nanomaterials-12-00813-f008:**
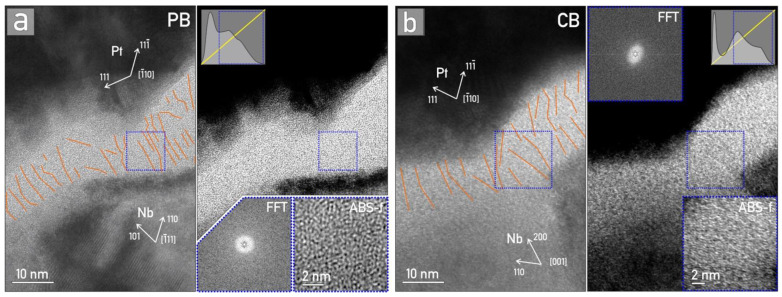
Conductive filaments (CFs) in anodic Nb memristors grown in phosphate (**a**) and citrate buffers (**b**).

## Data Availability

The data presented in this study are available on request from the corresponding author.

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
