# Peer review of "Impact of Electrolyte Incorporation in Anodized Niobium on Its Resistive Switching"

_nanomaterials, 2022, doi:10.3390/nano12050813_

Round 1

Reviewer 1 Report

In general, this work presents interesting data regarding the developing memristors based on niobium oxide. The proposed method of fabrication of memristors layers using anodization process looks promising and the developed devices demonstrated good performance and stability. This paper can be considered for publication after minor revision.

  1. In the introduction please comment more on the novelty of this work.
  2. 1a. Please comment if sample shown in this image is PB or CB sample? If it is PB sample, for example, the EDS line profile for CB sample is it similar?
  3. Fig 7a. Please provide more comments on observed irreversible change of resistance of CB sample.
  4. Line 413. The sentence reads. “of P and its atomic interactions may prevent the formation of a high number of fully-developed concurrent CFs.”. However, from Fig. 8 one can see that PB sample has higher density of CFs.
  5. Is there any difference in the samples shown in Fig. 8a and 8b (sample PB) and 8c and 8d (sample CB)? If those are images from the same samples in different places what is the reason to show those images?

Reviewer 2 Report

Manuscript investigate impact of electrolyte species incorporation in anodized niobium thin film on its resistive switching. Topic is interesting and novel, but few remarks are important. 
1) For analysis of electrolyte species incorporation two different electrolyte was used, and only one anodizing parameters was applied. At it is generally well known, the anodization parameters influence morphology and properties of anodic oxides, included here their chemical composition. As in presented manuscript was taken only one parameters of anodizing, in my opinion in the manuscript describe the influence of used to anodizing two different electrolyte, not investigate impact of incorporate spieces. 
Moreover, more details about anodization procedure should be given in Experimental part as well as in Introduction part, what is also connected with additional references needes. 
2) To calculate relative concentration of Nb, O and Pt the Cliff-Lorimer's ratio method was used. Since the thickness of prepaerd materials was around 250 nm why this method for calculation was selected? Generally, Cilff-Lorimer's method is recommended for samples with thickness of about 10-20 nm.
3) The carrefull checking of reference is required. I.e. in ref the name of authors are missing.

Reviewer 3 Report

  1. For better representation, description and comparison of the behavior and properties of the fabricated niobium oxide memristors, correspondent memristor models could be applied. Useful information could be found in the newest papers and books on memristor models with activation thresholds and improved window functions, for example:
    1. Mladenov V. Analysis of Memory Matrices with HfO2 Memristors in a PSpice Environment. Electronics. 2019; 8(4):383. https://doi.org/10.3390/electronics8040383.
    2. Mladenov V. A Unified and Open LTSPICE Memristor Model Library. Electronics. 2021; 10(13):1594. https://doi.org/10.3390/electronics10131594
  2. Try to include several equations, related to the described memristor elements, their behavior and modeling.
  3. In the end of the Introduction, a brief description of the following sections could be presented, for example “Section 2 presents …, … in section 3 is shown …”.
  4. The key-word “niobium oxide” must be included in the paper text.
